# Mixing Cells for Vascularized Kidney Regeneration

**DOI:** 10.3390/cells10051119

**Published:** 2021-05-06

**Authors:** Michael Namestnikov, Oren Pleniceanu, Benjamin Dekel

**Affiliations:** 1Pediatric Stem Cell Research Institute, Edmond and Lily Safra Children’s Hospital, Sheba Medical Center, Tel Hashomer, Ramat Gan 52621, Israel; michaeln@mail.tau.ac.il; 2Sackler Faculty of Medicine, Tel Aviv University, Ramat Aviv, Tel Aviv 69978, Israel; orenplen@tauex.tau.ac.il; 3The Kidney Research Lab, Institute of Nephrology and Hypertension, Sheba Medical Center, Tel Hashomer, Ramat Gan 52621, Israel; 4ediatric Nephrology Division, Edmond and Lily Safra Children’s Hospital, Sheba Medical Center, Tel Hashomer, Ramat Gan 52621, Israel; michaeln@mail.tau.ac.il

**Keywords:** kidney regeneration, vascularization, cellular therapy, stem-cells, iPSCs organoids

## Abstract

The worldwide rise in prevalence of chronic kidney disease (CKD) demands innovative bio-medical solutions for millions of kidney patients. Kidney regenerative medicine aims to replenish tissue which is lost due to a common pathological pathway of fibrosis/inflammation and rejuvenate remaining tissue to maintain sufficient kidney function. To this end, cellular therapy strategies devised so far utilize kidney tissue-forming cells (KTFCs) from various cell sources, fetal, adult, and pluripotent stem-cells (PSCs). However, to increase engraftment and potency of the transplanted cells in a harsh hypoxic diseased environment, it is of importance to co-transplant KTFCs with vessel forming cells (VFCs). VFCs, consisting of endothelial cells (ECs) and mesenchymal stem-cells (MSCs), synergize to generate stable blood vessels, facilitating the vascularization of self-organizing KTFCs into renovascular units. In this paper, we review the different sources of KTFCs and VFCs which can be mixed, and report recent advances made in the field of kidney regeneration with emphasis on generation of vascularized kidney tissue by cell transplantation.

## 1. Introduction

The mammalian kidney is an essential organ, carrying out numerous functions such as blood filtration, regulation of water and solute balance and hormone production.

Chronic kidney disease (CKD) is a progressive decline in kidney function, which can deteriorate to the state of end stage kidney disease (ESKD). Deterioration of kidney function measured by glomerular filtration rate (GFR) is accompanied by tubular interstitial fibrosis, a typical manifestation of CKD, loss of nephron cells and reduction of nephron number. The only options available for ESKD patients are dialysis and/or lifesaving kidney transplant [1]. Dialysis highly diminishes the patient’s quality of life and is associated with multiple co-morbidities [2]. In comparison, kidney transplant is considered a superior solution; however, due to the paucity of available donor organs and the requirement for lifelong immunosuppressive therapy, which in itself may result in a myriad of side effects [3], it is still not ideal. Consequently, medical health care systems and patients worldwide are yearning for a solution to CKD, and kidney regenerative medicine possesses great promise.

The field or regenerative medicine utilizes cells, small molecules, and extra-cellular matrices to reconstitute the native tissues function [4]. The challenge of kidney regeneration is tremendous–several crucial and complex microanatomical structures must been reconstituted: (1) the nephron, the functional subunit of the kidney, which is further divided into specialized segments, each with its own repertoire of transporters responsible for specific absorption and secretion of proteins, ions, and waste products; (2) the collecting duct system, which diverts filtered urine from the nephrons to the renal pelvis which funnels it to the ureter and the rest of the urogenital system; (3) an intricate microvasculature which serves as an interface between the systemic blood and the nephrons, allowing transport of solutes, concentration of urine via counter current mechanism and exerting hormones responsible for blood pressure and red cell production; (4) renal interstitium, composed of various cell types (e.g., fibroblasts) and extra-cellular matrix (ECM), providing the structural framework for the kidney.

Since progression of CKD is characterized by loss of kidney cells and nephrons, it would be beneficial to find a cell source which would replenish lost cells and halt disease progression, or perhaps reverse it. Multiple cell sources have been suggested for kidney regeneration in the past: fetal kidney, adult kidney, and urine derived kidney cells. Furthermore, in the past decade, protocols for generating pluripotent stem cell-derived kidney organoids (PSC-KOs) have been devised which allow the creation of autologous renal tissue [5]. However, to comprehensively combat CKD, one must also consider the effect the disease has on the microvasculature. Peritubular capillary rarefaction is a notable characteristic and driving force of CKD [6]. Destructive alterations in the kidney structure causing initial hypoxia on nephrons and interstitial capillaries lead to a vicious cycle which further aggravates scarring and diminishes kidney function [7]. This suggests that any holistic cellular regenerative therapy must incorporate the regeneration of the microvasculature to support KTFCs in a hostile, hypoxic, and partially fibrotic environment. Failure to provide vascular support to the engrafted cells could lead them to be subjected to a similar fate as the host’s paranchyma [8]. Furthermore, it cannot be expected that the host’s vasculature will provide ample blood flow to the transplanted cells since it is already failing to do so with the native kidney tissue.

The aim of this review is to concisely summarize the various cell sources for KTFCs and VFCs used for regenerating nephron parts and vascular components of the kidney, report on the most recent advances in cell-based methods of vascularized kidney regeneration and describe the remaining challenges in the field.

## 2. Sources for Kidney Tissue Forming Cells (KTFCs)

### 2.1. Fetal Kidney (FK)

Various studies in the past two decades have uncovered the complex and unique process of nephrogenesis, which culminates in the formation of a functional kidney [9,10].

The kidney tubules, representing the epithelial compartment, develops via reciprocal interaction of the metanephric mesenchyme (MM) and the ureteric bud (UB) [11]. The invading UB causes the MM to condense and differentiate into pre-tubular aggregates, coma shapes and s-shaped bodies, nascent nephron structures, which further mature into segmented renal tubules [12]. The SIX2+ cells are established as the stem cells residing in the MM, differentiating into all segments of the nephron: podocytes, proximal tubule, loop of Henle, distal tubules, and connecting tubules [13] (Figure 1a). The UB, in contrast, gives rise to the collecting duct and more distal structures. It is currently accepted that the adult kidney is devoid of “professional” stem cells, which makes the fetal kidney the main go-to source for kidney regeneration [14]. Our group and others have identified cell surface markers which can be used to isolate SIX2+ nephron progenitor cells (NPCs) for expansion and transplantation in kidney regenerative medicine [15,16,17,18] (Figure 1a). Defined culture mediums were devised for 2D and 3D expansion of isolated NPCs. Brown et al. introduced a nephron progenitor expansion medium (NPEM) to culture magnetically sorted NPCs in 2D [19]. Li et al. have developed a long-term culture system for mouse and human NPCs as 3D aggregates [20]. Transplanted human NPCs in mouse omentum form a tubular structure with distinct proximal (LTL) and distal (DBA) markers.

The renal interstitium compartment, shown to originate from FOXD1+ progenitors, residing within the early MM plays a vital role in the differentiation of nephron progenitors [21]. Even though interstitial cells are also charged with production of renin and erythropoietin and play a part in kidney interstitial fibrosis in disease [22], this compartment is often overlooked and is only now starting to receive proper attention [23,24].

The vasculature is the third critical compartment of the kidney in both health and disease. The renal vasculature can be divided into major blood vessels (e.g., renal artery and vein, afferent and efferent arterioles, etc.), glomerular vasculature and the peritubular capillaries (vasa recta). The origin of these compartments in embryonic kidney development is still under debate, and two main theories exist as to its origin. The first theory argues that vascularization is an extra-renal process driven by angiogenesis–pre-existing vessels sprout new vessels which penetrate the MM and vascularize the vascular cleft of the proximal region of nascent nephrons during the s-shape phase. The second theory argues that vascularization is an intra-renal process driven by vasculo-genesis, in which resident epithelial stem cells (also known as angioblasts) differentiate and form blood vessels around the developing nephrons. This process is thought to be driven by the secretion of VEGFA by the podocyte progenitors in the vascular cleft in the proximal s-shaped body [25]. Recent research in the field has of fetal development has attributed vascularization mainly to a combination of angiogenesis and vascularization, or to angiogenesis alone [26]. Dekel et al. found that the grafts are mostly vascularized by the host animal. Staining of pig MM xenografts transplanted into the rat renal subcapsular space showed only rat-positive endothelial cells vascularizing the grafts [27]. Since the angioblasts surround the MM and UB, it is possible that donor renal vasculature was not present after transplantation, since these niches were not picked up. Whole mount tracking of CD-1 fetal mouse kidney vascularization failed to trace CD31-cells forming de novo vessels at the periphery of the kidney, and only CD31+ endothelia was present which could be traced back to the renal artery [28]. Transplantation studies of developing FKs show supporting evidence of donor derived stromal angioblast driven vasculo-genesis, although it is almost always accompanied by the host’s blood vessels [29,30,31]. PECAM-1, a marker of mature endothelium, is expressed in E11.5 methanephroi before markers (Flk-1, Flt-1 and Tie-2) of angioblasts are detectable [32].

Theoretically, the fetal kidney is the ideal source for regeneration, since by definition, it harbors all progenitor populations needed to generate a kidney. Indeed, human FK precursors have been transplanted to form miniature kidneys that generate dilute urine [33]. However, one caveat to this strategy is that, if indeed fetal vascularization requires extra-renal cell types, then proper vascularization cannot be obtained by relying on FK alone.

### 2.2. Adult Kidney (AK)

Autologous adult renal tissue is an attractive cell source since it does not have the same ethical limitations as the FK and, more importantly, potentially allows for autologous cell therapy. The adult kidney harbors heterogenous lineage restricted precursor cells (LRPCs) which can clonally expand to replace damaged cells after injury [34]. These LRPCs allow the generation of AK organoids, also known as tubuloids [35] (Figure 1b). Tubular fragments embedded in ECM, grown in Wnt amplifying medium (R-SPO-1 conditioned medium), allow the long-term propagation of segment specific tubuloids from AK tissue and urine samples. AK organoids can overcome one of the major hurdles of using 2D culture when propagating highly passaged primary epithelial cells, i.e., loss of phenotype by undesirable epithelial to mesenchymal transition [36]. This technology has the potential to achieve the critical mass of cells needed for kidney regenerative medicine. AK tubuloids, however, have not yet been used in the context of kidney regeneration. To that end, another solution, which has been demonstrated by our group, is to expand AK cells in 3D as nephrospheres (nSPH) [37,38]. Once transplanted to the kidney capsule of mice, these nSPH self-assemble into renal tubules which partially integrate to the host nephrons, attract mouse blood vessels and improve renal function in CKD mice. Another exciting AK cell source which does not require an invasive procedure is the collection of urinary kidney cells found in the urine of patients. Their potential to regenerate the kidney is still investigated; however, they are an accessible somatic cell source for reprograming and differentiating into various tissues [39,40], including the kidney [41].

### 2.3. Pluripotent Stem Cell Derived Kidney Organoids

In the past decade, multiple protocols for generating kidney organoids from pluripotent stem cells (PSCs) have been developed [42,43,44,45]. The main strategy is to mimic the derivation of the MM and UB from the intermediate mesoderm (IM) to form convoluted organoids which include all the segments of the nephron and collecting ducts (Figure 1a). Protocols for derivation of PSC-KOs are varied, but usually consist of inducing differentiation of embryonic stem-cells or induced pluripotent stem cells (iPSCs) by introduction of CHIR99021 and FGF9, subsequently embedding the primed cells into Matrigel or suspending in 3D [46] (Table 1).

As in the FK, protocols for isolation and expansion of NPCs from PSC-KOs have been devised [47]. Single-cell analysis of well-established protocols showed presence of a small endogenous pool of endothelial progenitor cells (EPCs) in PSC-KOs [48], making them a candidate for a one stop solution for kidney regeneration, as an autologous cell source with accompanying EPCs which can facilitate vascularization. The shortcomings of this cell source are the high batch-to-batch variability [49] and immature nature of incompletely differentiated cells [50]. In vitro, even though EPCs are derived during differentiation of PSC-KOs, they fail to differentiate into functional vasculature and rarely invade the glomeruli [26]. When transplanted in mice, PSC-KOs vascularize extensively by host and graft functional blood vessels [51]. Furthermore, improved maturation of glomeruli is evident by the formation of glomerular basement membrane (GBM), fenestrated endothelium, podocyte foot processes and Bowman’s capsule. In vivo, PSC-KOs promote angiogenesis through production of VEGF, which promotes viability of graft, vascularization, and maturation [52]. Addition of VEGF to in vitro cultures did not improve vascularization of glomeruli but increased the pool of ECs in the organoids [53]. A possible solution to the problem was presented by Homan et al., when kidney organoids were grown in microfluidic chips under the mechanical cues of fluid shear stress. This resulted in increased expansion of ECs, enhanced maturation of podocytes and invasion of CD31+ vascular structures into podocyte clusters [54]. Of note, addition of VEGF disrupted vascularization of LTL+ tubules, similarly to the the effect of addition of VEGF inhibitor, indicating the importance of endogenous VEGF gradients for endothelial-epithelial communication. Further research is needed to compare the ability of perfused organoids with improved vascularization to connect to the hosts vasculature and nephrons.

## 3. Sources for Vessel Forming Cells (VFCs)

When cells/tissues are transplanted into a host they must be provided with oxygenation, nutrient delivery, and waste product removal. This can be accomplished via angiogenic processes in which the host’s vasculature connects or surrounds the transplanted cells/tissues. However, evidence shows that without support of encircling smooth muscles (also known as pericytes) these blood vessels are short lived [55]. This means that on top of cells which promote de-novo vasculo-genesis, such as ECs, cells which stabilize vessels once they are formed are also required. MSCs from autologous adipose tissue can support the generation and stabilization of vascular tissue by functioning as perivascular cells and promote tissue regeneration via paracrine mechanisms [56]. Hence, most approaches for tissue engineering and regenerative medicine of vascularized tissues use ECs and MSCs synergistically [57] (Figure 1c). A popular cell source for ECs are human umbilical vascular endothelial cells, HUVECS. Unfortunately, these VFCs can be propagated for a very limited number of passages and cannot be used as an autologous source. A promising cell source for autologous human ECs is the peripheral blood. These endothelial progenitor cells, commonly known as endothelial colony-forming cells (ECFCs), can be harvested non-invasively and differentiated into proper ECs [58]. Together with supporting cells, smooth muscles cells or MSCs, they can form dense microvascular networks [59,60]. The ultimate autologous cell source which is not riddled with ethical concerns are iPSC derived ECs. Somatic cells such as skin fibroblasts can be reprogrammed to into iPSCs and differentiated in ECs [61]. Upon transplantation into ischemic tissue, they improve local blood flow by connecting to the vasculature [62]. Recently, Cho et al. have shown that co-injection of human retinal endothelial cells, together with iPSC derived ECs, formed robust vascular networks which integrates to the host’s vasculature in an oxygen-induced retinopathy mouse model [63].

It is also important to consider the resident vascular and stromal cells which are present in primary fetal and adult tissues, which might also be sorted out, propagated, and utilized. The kidney vasculature has a distinct transcriptional profile [64,65] which is also highly heterogenous [66], partly due to the specialized glomerular vasculature, thus it is safe to assume that the regenerative capability of the vascular compartment of the kidney is partly regulated in a tissue specific manner. This is why the niches of endothelial and stromal cells which exist in differentiation protocols of PSC-Kos [67] are primed to interact with the nephrogenic compartment by invading the glomerulus, and an addition of mechanical and chemical cues could be provided to generate extensive vascular networks, connecting glomerular-like structures to host’s circulation.

## 4. Mixing and Injection of KTFCs with VFCs and MSCs to Form Vascularized Kidney Tissue

For regenerative capacity testing, KTFCs and VFCs are often mixed in ECM hydrogels such as Matrigel and injected, usually into the kidney (Table 2). There are three main locations for cell transplantation in the kidney: subcapsular, intra-parenchymal and intra-arterial. Subcapsular injection of cells has multiple advantages–injected cells are easily contained under the kidney capsule, which is a highly vascularized site, and relatively easy to image and biopsy [68]. This strategy allows the vascularization of grafts and is mainly useful for exerting paracrine effects upon the injured host’s kidney. Intra-parenchymal injection of cells enables them to integrate into existing nephrons and vasculature; however, this is more harmful to the existing micro-architecture than subcapsular implantation [38]. Finally, intra-arterial injection is least harmful but also least effective for integration. In our experience, intra-arterial injection of KTFCs and VFCs allows successful engraftment but poor self-organization of kidney structures and vessels [69]. Similar results were shown in cardiac regeneration, where bone marrow mononuclear cells (MNCs) are lost in circulation when injected systemically [70]. Other groups have utilized intra-arterial injection of MSCs and achieved kidney engraftment of cells [71,72]; however, the efficacy of this method for delivering mixed cells which self-organize into functional tissue is still open to debate.

When mixing fetal KTFCs with VFCs to form vascularized kidney tissue, an intuitive strategy is to use resident VFCs from the native FK Mice fetal kidney organoids consisting of isolated endothelial precursors (CD31+ and/or Flk1+) mixed with isolated MM cells (Itga8+, Pdgfra−) to form kidney organoids which upon transplantation vascularize, mainly by the donor [73]. Interestingly, no difference was evident in vascularization when the VFC fraction was excluded, presumably due to compensatory angiogenesis by the host or stowaway endothelial precursors remaining in the organoids.

Our group has recently shown that co-administration of human AK cells, with ECFCs and MSCs, increases the tubulogenic capacity of injected kidney cells, upregulates self-renewal and mesenchymal to epithelial transition markers and facilitates connection to the host’s vasculature [69]. Although these “renovascular units” are an important advance for improving survivability and function of the kidney cells, they still lack the glomerular filtration component, thus beneficial effects are mainly attributed to paracrine effects and presumably secretion and reabsorption of solutes. Further studies will have to address the capabilities of renovascular units to ameliorate renal function in CKD mouse models.

Taniguchi’s group reported an ex-vivo method for generating vascularized organ buds by combining pluripotent stem cell-derived tissue progenitors (from various organs, including kidney), as well as tissue fragments, with ECs and MSCs. Upon intracranial transplantation in mice, the organ buds successfully integrated into the host [74]. Notably, and similar to our observations, the authors found that fetal renal progenitors transplanted in isolation (i.e., not within organ buds) failed to vascularize.

Sharmin et al. transplanted iPSC-derived kidney organoids with HUVECS and MSCs to the mouse kidney capsule [75]. After 20 days, host blood vessels integrated to the glomeruli, but HUVEC-derived blood vessels did not, suggesting that HUVECs are not able to integrate with glomeruli. Recently, a novel mixing protocol was devised: an asynchronous approach for generating kidney organoids. In this approach, cells at various time point of differentiation were mixed to enhance nephrogenesis and form a more complex network of ECs in vitro [76]. Mixed engrafted organoids vascularized well in-vivo, with SMA+ vascular networks invading PODXL+ clusters and CD31+ cells connecting to a WT1+ presumptive glomeruli.

## 5. Conclusions and Future Perspectives

The first approaches to kidney regenerative medicine utilized the embryonic kidney stem cell compartment in the FK, the MM, which gives rise to vascularized nephrons. Vascularization is a key component which will have to be addressed in order to allow regenerated kidney parenchyma to produce and secrete urine. Blood vessels need to invade the glomerulus to establish blood filtration and surround the other segments of the nephron to allow additional fine tuning of solute transport. It is important to note that the various KTFCs reviewed in this paper have different regenerative capacities: KTFCs from FK and PSC-derived kidney organoids can generate the entire nephron, while KTFCs from AK are lineage restricted and produce segment specific tubules. This constraint must be taken into consideration when planning vascularization and co-engraftment with VFCs and supporting cells. KTFCs from AK will not be able to reconstitute glomerular filtration since glomeruli are usually lost during culture. Nevertheless, the vascularized tubular structures created might add a layer of functionality to CKD patients via paracrine effects and absorption and secretion of solutes.

KTFCs from “immature” sources such as the fetal kidney and PSC-KOs promote host vascularization via VEGF secretion and endogenous VFCs. Most importantly, engrafted organoids exhibit blood vessel invasion to the glomeruli, a landmark of functional renal tissue. The AK cell source is purely epithelial and thus benefits from co-administration with VFCs, showing increased tubulogenic capacity. In case of transplantation of KTFCs from any source into CKD kidneys it is reasonable to consider that vascular dysfunction occurring in CKD would dampen host vascularization and hence mixing cells to achieve donor vascularization is likely to be required. The ultimate strategy for generating renovascular units, with glomerular blood filtration and vasa recta, is to engraft KTFCs with VFCs, which could be derived by differentiating PSCs into a vascularized kidney organoids in-vitro or in-vivo. To this end, further priming for vascularization could be addressed by providing mechanical cues and growth factor gradients promoting VFC proliferation and differentiation. This could lead to the development of a universal off the shelf kidney regenerative cell source for mixing KTFCs and VFCs, especially with the ability to modulate immunogenicity via CRISPR-CAS9 gene editing [77,78].

## Figures and Tables

**Figure 1 cells-10-01119-f001:**
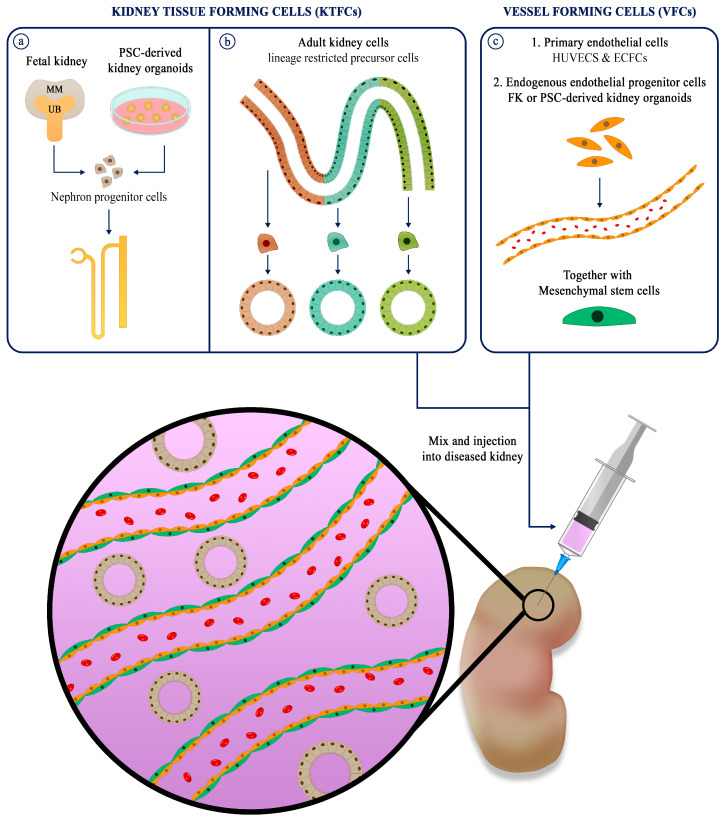
Overview of engraftment strategy for vascularized kidney regenerative medicine. (**a**) KTFCs from fetal kidney or PSC-derived kidney organoids have the capacity to differentiate into nephrons with glomerulus, proximal tubule, loop of Henle and distal tubule. (**b**) KTFCs from adult kidney derived from biopsies of urine samples are lineage restricted precursor cells, able to generate segment specific tubules. (**c**) VFCs such as endothelial cells and peritubular support cells (pericytes and mesenchymal stem cells) are added to the cell mix to support blood vessel formation. Adapted from reference [9].

**Table 1 cells-10-01119-t001:** Overview of well-established protocols for generation PSC-KOs. The cell source, differentiation factors and culture methods are summarized and presented for each protocol.

Pluripotent Stem-Cell Source	Factors Used in Differentiation	Culture Method	Reference
Human embryonic stem-cells and human iPSCs	Y27632, CHIR99201, B27	Matrigel sandwich spheroids	Freedman et al. [42]
Human embryonic stem-cells and human iPSCs	CHIR99201, Noggin, Activin A, FGF9	96-well, low-attachment	Morizane et al. [43]
Mouse embryonic stem-cells and human iPSCs	Y27632, BMP4, FGF2, Activin A, CHIR99021, Retinoic Acid, FGF9	Transwell membrane	Taguchi et al. [44]
human iPSCs	CHIR99201, FGF9	Transwell membrane	Takasato et al. [45]

**Table 2 cells-10-01119-t002:** Summary of studies mixing KTFCs with VFCs. KTFCs from fetal, adult kidneys and iPSC-derived kidney organoids. Primary VFCs used such as HUVECs, ECFCs, endogenous ECs from the developing fetal kidney and iPSC-derived organoids and MSCs.

KTFCs	VFCs	Supporting Cells	Area of Injection	Renal Compartment	Vessel Compartment	Reference
E11.5 mouse embryonic kidney Itga8+/Pdgfra− nephron progenitors	E11.5 mouse embryonic kidney CD31+ and/or Flk1+ endothelial cells	None	Kidney sub-capsule	Glomeruli and renal tubules	Donor and host arteriole network and glomerular capillaries	Murakami et al. [73]
E13.5 mouse embryonic kidney	HUVECS	MSCs	cranium	Glomeruli and proximal tubules	Glomerulus integrated with recipient circulation	Takebe et al. [74]
Adult and fetal kidney	Endothelial colony forming cells isolated from umbilical cord blood	MSCs	Kidney sub-capsule, parenchyma and intra-arterial	Proximal and distal lineage	Donor derived blood vessels	Pleniceanu et al. [69]
nephron progenitors from iPSC-derived kidney organoids	HUVECS	MSCs	kidney capsule specialized rod assisted engraftment	glomeruli, proximal and distal tubules	Host derived ECs integration to glomeruli	Sharmin et al. [75]
iPSC-derived kidney organoids	endogenous ECs from heterochronic mixing	None	kidney sub-capsule	glomeruli, proximal and distal tubules	Host derived ECs integration to glomeruli	Gupta et al. [76]

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
