# Peer review of "Mixing Cells for Vascularized Kidney Regeneration"

_cells, 2021, doi:10.3390/cells10051119_

Round 1

Reviewer 1 Report

General comments:

The authors emphasize the importance of the kidney and the need for cell therapy for chronic kidney disease (CKD). They describe fetal kidney (FK), adult kidney (AK), and PSC-derived kidney-organoids (PSC-KOs) as a source of cell therapy that the advantages and hurdles of each are discussed by referring to previous articles. Also, it explains the vessel-forming cells (VFCs) that can support them. However, the manuscript does not provide conclusive evidence for the hypothesis, and there are major improvements required to be advisable for acceptance.

1) The authors are talking about cell therapy for CKD, but information on CKD insufficient. It is suggested to include additional information about molecular or signal change for CKD at the beginning of the introduction.

2) The authors describe the differentiation type of metanephric mesenchyme (MM) and the ureteric bud (UB) (line 85-88). If figures about kidney structure exist in the manuscript, readers will be easier to understand. Therefore, it is recommended to add a figure.

3) The subtitles for the fetal kidney and adult kidney parts in the text are abbreviated, but the subtitles for the PSC-derived kidney organoid parts are not (line 152). Like other subtitles, I ask you to write an abbreviation of PSC-derived kidney organoids (PSC-KOs). In addition, PSC-Kos should be change PSC-KOs (line 157).

4) The authors describe PSC-derived kidney organoids, which say that the protocol of PSC-KO was developed (line 153-177). However, I think that information on PSC-KOs is over-abbreviated. So, I strongly suggest explaining the mechanism of signal or material (cytokine and chemical) that are important in the differentiation of cells that make up organoids. Furthermore, previous article protocols for PSC-KOs ([39]-[42]) would be nice to simply make a table like Table 1.

5) If the RFTCs in Figure 1 is a typo, correct them with RTFCs (line 70).

Author Response

We thank the referee for his time and effort to make this review better. 

  1. We have added additional info about CKD pathology - line 30 "Deterioration of CKD..." and line 59 "Peritubular capillary rarefaction"
  2. We have added the MM and UB compartments in figure 1 at the fetal kidney section
  3. We have changed all pluripotent stem cell derived organoids to PSC-KOs, typo at line 157 was fixed.
  4. Additional information about factors used for the derivation of PSC-KOs was added in line 159 and a table was added containing a summary and comparison between the well-established protocols.
  5. Typo was fixed.

Reviewer 2 Report

This review examines most of the major issues surrounding the development of tissue engineered/regenerative medicine replacement of kidney function. Although the development of clinically useful engineered kidney replacement is still a long way off, this review provides a solid foundation for the next level of research efforts.

There are a very few typographical errors (Table 1, arteriole; line 206 needs an edit)

Author Response

We thank the referee for comments.

We have fixed the typo.

Reviewer 3 Report

This is very importan literature review aimed to present a current status of kidney regenerative medicine and describes the potential use of different cell sources for kidney regeneration and specially, for kidney vascularization. 

The authors cited paramount references and the manuscript is well wrote.

Author Response

We thank the referee for his supporting comments.